**Data Availability Statement:** All relevant data are within the manuscript and its Supporting information files.

## RESEARCH ARTICLE

# Immunoinformatics-guided recombinant polypeptide-based enzyme-linked immunosorbent assay for seromonitoring of laboratory animals for minute virus of mice and Kilham rat virus

Charanpreet Kaur[1], Kandala Pavan Asrith[1], S. G. Ramachandra[2], Nagendra R. Hegde[1]*

1 National Institute of Animal Biotechnology, Hyderabad, India, 2 Indian Institute of Science, Bangalore, India

* hegde@niab.org.in

## Abstract

Subclinical infection of laboratory animals with one or more of several pathogens affects the results of experiments on animals. Monitoring the health of laboratory animals encompasses routine surveillance for pathogens, including several viruses. This study aimed to explore the development of an alternative assay to the existing ones for detecting infection of mice and rats with the parvoviruses minute virus of mice (MVM) and Kilham rat virus (KRV), respectively. Full-length VP2 and NS1 proteins of these parvoviruses, besides fragments containing multiple predicted epitopes stitched together, were studied for serological detection. The optimal dilution of full-length proteins and antigenic regions containing predicted epitopes for coating, test sera, and conjugate was determined using a checkerboard titration at each step. The assays were evaluated vis-à-vis commercially available ELISA kits. The results showed that an engineered fusion of fragments containing multiple predicted MVM VP2 and NS1 epitopes was better than either of the full-length proteins for detecting antibodies in 90% of the tested sera samples. For KRV ELISA, full-length VP2 was better compared to other individual recombinant protein fragments or combinations thereof for the detection of antibodies in sera. This report is the first description of an ELISA for KRV and an improved assay for MVM. Importantly, our assays could be exploited with small volumes of sera. The results also demonstrate the utility of immunoinformatics-driven polypeptide engineering in the development of diagnostic assays and the potential to develop better tests for monitoring the health status of laboratory animals.

## Introduction

The surveillance of laboratory animals for exposure to microorganisms is critical for monitoring the health status of animals. It is a prerequisite for optimal performance of investigations in laboratory animal research. In addition, such surveillance is essential for the well-

**Funding:** This study was supported by a grant (BT/
PR15810/ADV/90/198/2015) from the Department
of Biotechnology, Ministry of Science and
Technology, Government of India, to NRH and
SGR. The funders had no role in study design, data
collection and analysis, decision to publish, or
preparation of the manuscript.

**Competing interests:** The authors have declared
that no competing interests exist.

being of the personnel who are in close contact with animals and consequently exposed to
specific zoonotic pathogens. The prevalence of various pathogens in mouse and rat colonies
has been well studied in the US and Canada [1–3], Europe [4–6], Australia [7], South Africa
[8], Brazil [9] and Argentina [10]. Apart from Japan [11, 12], South Korea [13, 14], and Tai-
wan [15], there are very few reports documenting the prevalence of laboratory animal path-
ogens in other Asian countries. We had earlier reported through a small study that the
prevalence of pathogens infecting laboratory animals in India is higher than in developed
countries and that mouse hepatitis virus (MHV), mouse parvovirus (MPV), and minute
virus of mice (MVM) were the most prevalent [16]. Prevalence of sialodacryoadenitis virus
(SDAV), rat parvovirus (RPV), and Kilham rat virus (KRV) in laboratory rats has also been
documented in several countries [6]. In general, MPV and MVM in mice, and KRV, RPV,
and Toolans H-1 virus in rats are the most prevalent but frequently underestimated parvovi-
rus infections in laboratory animals [17]. Natural infections with these pathogens are fre-
quently subclinical and asymptomatic, although clinical signs of KRV infections have been
reported in gestating rats [18]. On the other hand, parvoviruses can have substantial delete-
rious effects on the outcome of experiments due to their immunomodulatory effects [19–
23]. Parvoviruses are highly contagious, being transmitted through exposure to infectious
feces, urine, nasal secretions, and contaminated fomites. They are also considerably resistant
to environmental extremes, thus increasing the risk of their persistence and spread when
undiagnosed cases exist in a colony. Therefore, identifying the infected laboratory animals
is critical to minimize the influence of parvovirus infections on research conducted on
them.

Parvoviruses are small, non-enveloped, single-stranded DNA viruses with a genome size of
~5000 nucleotides encoding two main open reading frames (ORFs). The left ORF encodes the
well-conserved non-structural proteins, NS1 and NS2, involved in viral replication and regula-
tion of the capsid gene expression. The right ORF encodes three structural proteins: VP1 (viral
protein 1), VP2, and VP3. Alternative splicing of the same pre-mRNA forms VP1 and VP2
proteins. VP2 protein lacks the N-terminal region of the VP1 polypeptide [24], resulting in the
~64 kDa mature protein. In DNA-encapsulating virus particles, several VP2 protein molecules
undergo proteolytic cleavage, which removes approximately 25 residues at the N-terminus to
generate the VP3 protein [25–27].

Hemagglutination inhibition (HI), indirect fluorescence assays (IFA), and enzyme-linked
immunosorbent assay (ELISA) are the most common serodiagnostic tests for detecting parvo-
virus infections in mice and rats. The conventional methods employ purified inactivated virus
as the antigen [28, 29], and these are regularly used in commercial ELISA kits (XpressBio Life
Science, USA and Dynamimed, Spain). ELISA tests employing recombinant viral proteins
(VP2 and NS1 of MVM and KRV) have also been developed [30–33], and commercial kits
(Alpha Diagnostic International, USA, and Charles River Laboratory, USA) utilizing recombi-
nant VP2 or NS1 proteins are also available.

We set out to develop an indirect ELISA using both full-length VP2 and NS1 proteins of the
parvoviruses MVM and KRV as well as predicted immuno-dominant regions of these proteins
stitched together as antigens for detection of antibodies in sera of mice and rats, respectively.
We describe the cloning, expression, and purification of eight recombinant proteins: full-
length VP2, full-length NS1, a fragment of VP2 containing antigenic cluster, and two frag-
ments of NS1 antigenic clusters stitched together as one recombinant protein for both MVM
and KRV. These proteins were assessed for their ability to detect antibodies in ELISA using a
panel of mice and rat sera against MVM and KRV, respectively, compared to commercially
available kits.

## Material and methods

### Ethics statement

This study did not require ethical approval because it did not include human or animal trials. The serum samples used in this study were from an earlier study [16], where samples were sought from mice and rats maintained at several vivaria in different parts of India. These samples were collected by a veterinarian or trained personnel under a veterinarian's supervision. All the samples had been collected to monitor the health of mice and rats as part of routine surveillance following approval by the Institutional Animal Ethics Committee (IAEC), as per the guidelines laid down by the Committee for the Control and Supervision of Experiments on Animals, Department of Animal Husbandry and Dairying, Ministry of Fisheries, Animal Husbandry and Dairying, Government of India.

### Cell culture and propagation of viruses

The NIH-3T3 and C6 cells (American Type Culture Collection, USA) were propagated and maintained in Dulbecco's modified Eagle's medium (DMEM; Gibco, USA) supplemented with 10% heat-inactivated fetal bovine serum (FBS; Gibco), 100 U/mL of penicillin and 100 µg/mL of streptomycin (Invitrogen, USA). MVM and KRV (ATCC, USA) preparations were made by infecting cells (NIH-3T3 for MVM, C6 for KRV) and harvesting cell extracts after 48–72 h by three freeze-thaw cycles. The virus preparations were titred by infecting cells in 96-well plates, and the 50% tissue culture infective dose ($TCID_{50}$) was calculated by applying the Reed and Muench method [34].

### Prediction and screening of B-cell epitopes

The VP2 and NS1 protein sequences of MVM and KRV (GenBank Accession numbers J02275 and AF321230, respectively) were retrieved from the National Center for Biotechnology Information (NCBI; http://www.ncbi.nlm.nih.gov/). Linear B-cell epitope prediction was carried out using the BCPred B-cell epitope prediction server (http://ailab.ist.psu.edu/bcpred/) and the TMHMM server v.2.0 (for transmembrane helices in proteins). The length of the epitopes was restricted to 20 amino acids with specificity values of 90%. Based on the prediction results, protein fragments with good antigenicity were chosen and designated as A, C, D, E, F, and G (see Table 1).

### Construction of recombinant plasmids

Viral DNA was extracted from culture supernatants of MVM- and KRV-infected cells using the NucleoSpin Virus RNA/DNA Extraction kit (Macherey-Nagel, Germany). Primers (Table 2) (Integrated DNA Technologies, USA) were designed to amplify the full-length genes

**Table 1. Genes or fragments selected for amplification.**

| Fragment designation | Gene | Amino acid (aa) residue location in the gene | Polypeptide length (aa) |
|---|---|---|---|
| A | MVM *VP2* | 193–400 | 208 |
| C | MVM *NS1* | 309–485 | 177 |
| D | MVM *NS1* | 511–702 | 192 |
| E | KRV *VP2* | 309–580 | 272 |
| F | KRV *NS1* | 210–512 | 303 |
| G | KRV *NS1* | 553–653 | 101 |

**Table 2. Primers used in the study.**

| Gene(s) or fragment(s) | Primer designation | Sequence (5' to 3') | Product size (bp) |
|---|---|---|---|
| KRV *VP2* FL | VP2 KRV Nhe 51 | CATGTG**GCTAGC**ATGAGTAATGACGCCAATACC | 1773 |
| | VP2 KRV Hind 31 | GGCCGC**AAGCTT**TTAGTATGTGTTGCGAG | |
| KRV *NS1* FL | NS1 KRV BamH 51 | GGATCC**GGATCC**ATGGCTGGAAACGCTTACTCC | 2043 |
| | NS1 KRV Hind 31 | GCAGAT**AAGCTT**TTAGTCCAATGTCAGTGAATC | |
| MVM *VP2* FL | VP2 MVM Nhe 51 | CATGTG**GCTAGC**ATGAGTGATGGCACCAGCCAAC | 1788 |
| | VP2 MVM Xho 31 | GCAGAT**CTCGAG**TTAGTAAGTATTTC | |
| MVM *NS1* FL | NS1 MVM Nhe 51 | CATGTG**GCTAGC**ATGATAAGCGGTTCAGGGAG | 2190 |
| | NS1 MVM Hind 31 | GGCCGC**AAGCTT**TTAGTCCAAGTTCAGCGGC | |
| MVM *VP2* fragment A | VP2 MVM A Nhe51 | CATGTG**GCTAGC**AACTCAATGGAAACACTTG | 621 |
| | VP2 MVM A Hind31 | GGCCGG**AAGCTT**TTATTCATCCCATGTGTAGCG | |
| MVM *NS1* fragment C | NS1 MVM C Nhe51 | CATGTG**GCTAGC**ACGGTTGAAACCACAG | 528 |
| | NS1 MVM C BamH131 | TCCACC**GGATCC**GCCGTTGGTACAGTCATTAAATGG | |
| MVM *NS1* fragment D | NS1 MVM D BamH151 | AACGGC**GGATCC**GGTGGACAAACTATTCGCATT | 573 |
| | NS1 MVM D Hind31 | GGCCGG**AAGCTT**TTATCTCAAATCCTCCTCG | |
| KRV *VP2* fragment E | VP2 KRV E Nhe51 | CATGTG**GCTAGC**CAAGCACAGGCCAACAGATTTG | 813 |
| | VP2 KRV E Hind31 | GGCCGG**AAGCTT**TTAGTTGCGAGGTACAGG | |
| KRV *NS1* fragment F | NS1 KRV F BamH51 | CATGTG**GGATCC**TACTTTCTAAGCAAAAAGAAAATATGTACC | 906 |
| | NS1 KRV F Hind32 | TCCACC**AAGCTT**GCCGTTGAGCATTCTGTCTC | |
| KRV *NS1* fragment G | NS1 KRV G Hind51 | AACGGC**AAGCTT**GGTGGAAAATGGGGCAAAGTTCC | 300 |
| | NS1 KRV G Xho31 | CCAGAT**CTCGAG**TTATCTCAGATCCGCCTCG | |

Note: FL = full-length; bold and underlined sequences indicate restriction sites

and fragments containing clusters of antigenic epitopes. Full-length *VP2* and *NS1* genes of MVM and KRV were PCR amplified from the respective viral DNA using gene-specific primers (Table 2), along with desired restriction sites for cloning and the intent to obtain recombinant proteins with N-terminal hexahistidine tag. All PCR reactions were performed with 5 ng of template DNA, 200 μM dNTP, 0.2 μM forward primer, 0.2 μM reverse primer, 1X PCR buffer, and 1.25 units of PrimeSTAR GXL DNA Polymerase (Takara Bio Inc., Japan) in a 50 μL reaction. DNA was denatured at 95˚C/3 min and then amplified by 30 cycles of denaturation (95˚C/10 sec), annealing (55˚C/15 sec), and extension (68˚C at 1 min/kb fragment size) followed by a final amplification at 72˚C for 10 min. The amplified DNA was purified using the QiaQuick PCR purification kit (Qiagen, Netherlands). The purified DNA was digested with desired restriction enzymes (New England Biolabs, USA) and cloned into a similarly digested and dephosphorylated (with shrimp alkaline phosphatase; NEB, USA) pRSETA vector (ThermoFisher Scientific, USA) by using T4 DNA ligase (New England Biolabs, USA).

Fragments A (621 bp) and E (813 bp) were cloned into a pRSETA as described above. For the construction of the fragments CD (1164 bp) and FG (1335 bp), individual C (528 bp), D (573 bp), F (906 bp), and G (300 bp) fragments were PCR amplified, followed by gene splicing by overlap extension (SOE) to join the two coding sequences in frame to obtain CD and FG (Fig 1) respectively. For this, the purified fragments (C, D, F, and G) were mixed in equimolar concentration to obtain a final concentration of 1 ng/μl and used as a template for SOE-PCR before cloning CD and FG into the pRSETA vector. Joining was achieved by including nucleotides representing amino acids GGSGG and GKLGG as linkers, respectively, between C and D and F and G fragments. The plasmids were transformed into *E. coli* DH5α cells (Invitrogen,

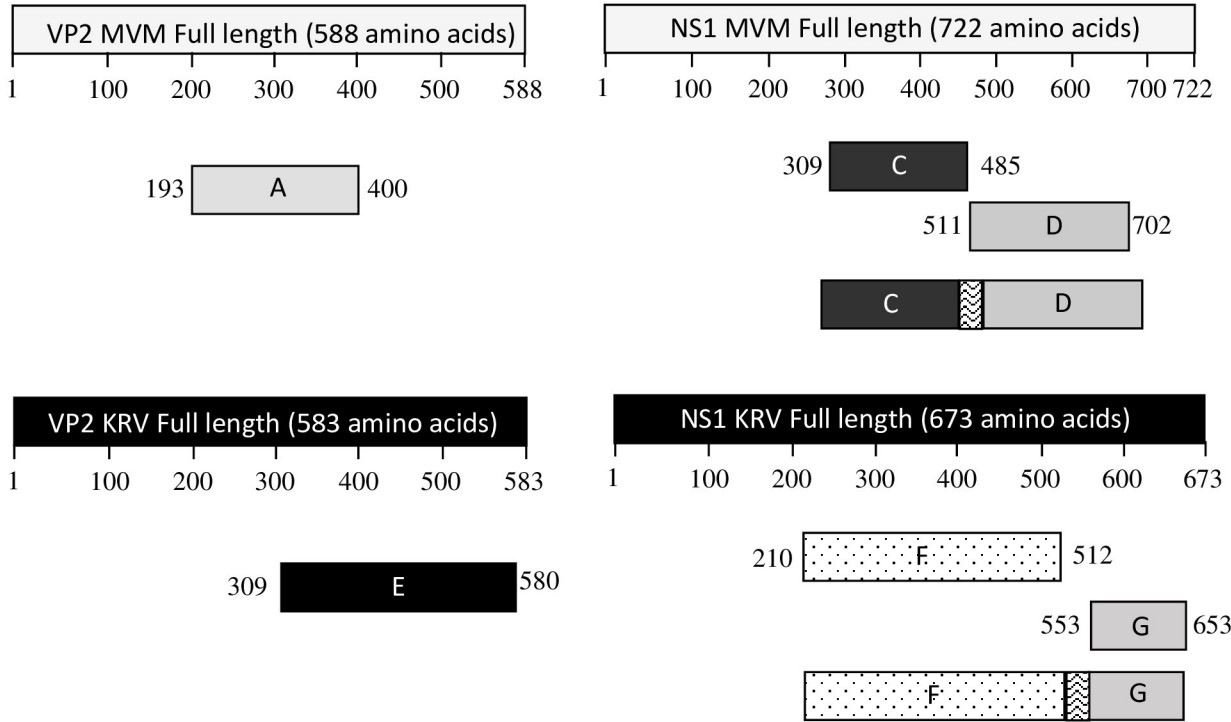

**Fig 1. Illustrative diagram of the constructs generated in this study.** All polypeptides were designed to contain a 6-His tag at the N-terminus. The linker between the fused fragments is five amino acids long.

USA), and colonies were screened by PCR for the presence of the insert. The recombinants were confirmed by sequencing the insert with T7P and T7tn primers. Recombinant plasmid DNAs were isolated from DH5α cells.

## Expression and purification of recombinant proteins

Recombinant plasmid DNAs were transformed into *E. coli* BL21 (λDE3) (Invitrogen, USA) and colonies were selected on MDAG plates containing 100 μg/ml ampicillin and 30 μg/ml chloramphenicol. Small-scale cultures were screened, and a clone that expressed the recombinant proteins maximally was selected for large-scale expression. Luria–Bertani medium (1 L) containing antibiotics was inoculated with a 100-fold dilution of overnight grown culture and incubated at 30˚C with shaking at 250 rpm. The culture was induced at 0.6–0.8 $OD_{600}$ nm with 1 mM final concentration of isopropyl-β-D-1-thiogalacto-pyranoside (IPTG; Sigma-Aldrich, USA) for 2 h, at 30˚C with shaking at 250 rpm. The culture was centrifuged, and the cell pellet was re-suspended in Tris loading buffer pH 7.5 (TLB; 50 mM Tris, pH 7.5, 500 mM NaCl), containing 200 μg/ml of lysozyme (ThermoFisher Scientific, USA), and 0.1 mM phenylmethyl-sulfonyl fluoride (PMSF; Sigma-Aldrich, USA), and incubated on ice for 30 min. Following sonication, the lysate was centrifuged at 20,000 *g* for 30 min at 4˚C. Soluble and insoluble fractions were analyzed by sodium dodecyl sulphate–polyacrylamide gel electrophoresis (SDS-PAGE) for the localization of recombinant proteins.

Purification was performed using 1 L of IPTG-induced culture biomass (~3.5 g wet weight) at 5–8˚C using an AKTA pure system (GE Amersham Health Sciences, Sweden). For insoluble fractions, the sonicated pellet was solubilized in TLBU (TLB containing 8 M urea). Soluble and

insoluble fractions were filtered through a 0.22 μ filter and loaded at 1 ml/min on a 1 ml HisTrap column (GE Healthcare Life Sciences, Sweden) equilibrated in TLBU. The column was washed with 20 ml of TLBU, and the bound protein was eluted with a 10 column volumes (CV) linear gradient of 0–500 mM imidazole in TLBU. The fractions were analyzed by SDS–PAGE. The fractions containing the desired protein were pooled and dialyzed against phosphate-buffered saline (PBS, pH 7.2). For the insoluble fractions, dialysis was carried out against progressively decreasing urea concentrations (6, 4, 2, 1, and 0 M) at 4˚C. The dialyzed protein was centrifuged at 30,000 $g$ for 1 h at 4˚C, and the concentration of the protein was estimated by the bicinchonicic acid (BCA) assay (ThermoFisher Scientific, USA) using bovine serum albumin (BSA; Sigma-Aldrich, USA) as the reference.

## Western blotting

The proteins were subjected to SDS–PAGE under reducing conditions and electro-blotted onto a polyvinylidene fluoride (PVDF) membrane (Bio-Rad Laboratories, USA). The membrane was blocked with 2% skimmed milk (ThermoFisher Scientific, USA) in PBS containing 0.05% Tween-20 (PBST) and washed three times for 10 min each, with PBST. Then, HRP-conjugated anti-His monoclonal antibody (Clontech Laboratories, USA) was added at 1:10,000 dilution in 1% skimmed milk in PBST (1% SMPBST) and incubated for 1 h at room temperature. After washing three times with PBST, and three washes with PBS, the membrane was developed with SuperSignal West Pico Substrate (Thermo Fisher Scientific, USA).

## Development of indirect ELISA

The checkerboard procedure was used to optimize the different parameters, including coating concentration, coating buffer (PBS, pH 7.2; carbonate-bicarbonate buffer, pH 9.6), coating temperature (4˚C, room temperature, 37˚C), coating time (overnight, 1 h, 2 h), blocking agent and concentration (skimmed milk, bovine serum albumin), and the dilutions of primary and secondary antibodies. Once standardized, Maxisorp plate (Nunc-ThermoFisher Scientific, USA) wells were coated with 100 μl of protein (5, 2.5, 1, or 0 μg/ml in 50 mM carbonate-bicarbonate buffer, pH 9.6) or BSA as the control protein, and incubated for 2 h at 37˚C. Plates were washed thrice with PBST and blocked with 300 μl/well of 5% SMPBST for 1 h at 37˚C. Then, different dilutions (0, 1:25, 1:50, 1:100, 1:200) of positive and negative sera were added, followed by incubation at 37˚C for 1 h.. After washing the wells with PBST, 100 μl/well of HRP-conjugated goat anti-mouse or anti-rat IgG (H+L) (Jackson ImmunoResearch Laboratories, USA) diluted 1:10,000 in 1% SMPBST was added and incubated for 1 h at 37˚C. The wells were washed again with PBST, followed by three washes with PBS. The wells were reacted with 100 μl of tetramethylene benzidine (TMB) (ThermoFisher Scientific, USA) for 15 min, and the reaction was terminated with 100 μl of 1N $H_2SO_4$. The color was read at 450 nm using a microplate reader. In all the assays, negative and positive control sera and no coating and coating with BSA, were set up on each plate.

## Statistical analysis

Statistical analysis was carried out using GraphPad Prism version 8.0.2. Each experiment was independently performed in duplicate or triplicate to ensure reproducibility. Data are shown as mean ± SD. Statistical significance was calculated by paired, two-tailed Student's t-test for two-group analysis. A p-value $< 0.05$ was considered significant.

## Results

By applying immuno-informatics, epitopes potentially present on the surface of proteins were targeted, and the most promising regions representing potential short linear epitopes were identified (Table 1). In addition, segments of proteins containing several overlapping and non-overlapping but closely located epitopes were identified. Potential antigenic regions C, D, F, and G were cloned by SOE-PCR to obtain CD and FG proteins with a five amino acid spacer between them. These fusion proteins and the full-length recombinant antigens (VP2, NS1 of

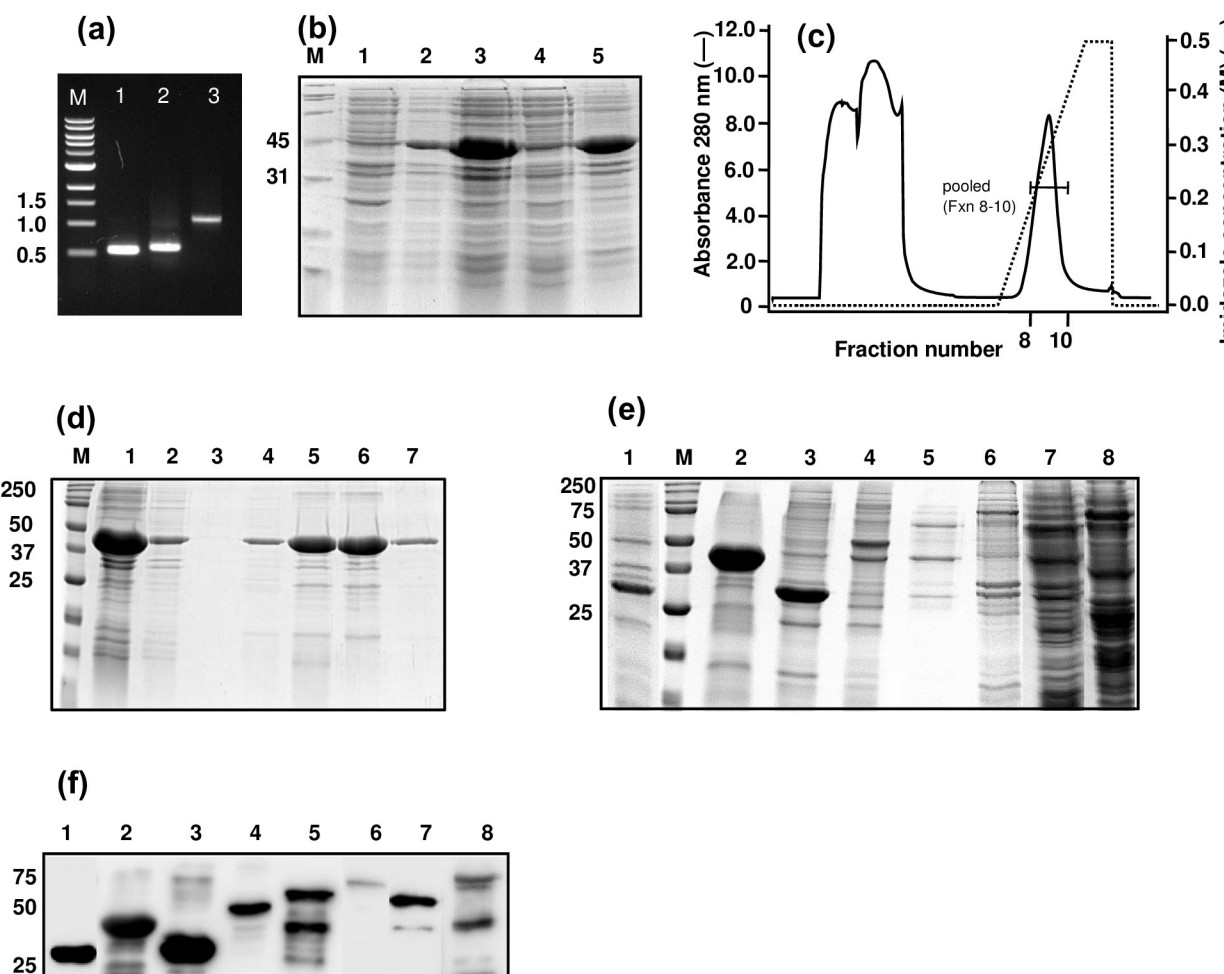

**Fig 2. Cloning, expression, and purification of the CD polypeptide. (a) Cloning of CD.** Fragments of C and D were amplified separately and then joined by SOE-PCR. M, DNA marker; 1, PCR amplified fragment C (528 bp); 2, PCR amplified fragment D (573 bp); 3, joined CD fragment (1.1 kb). **(b) SDS-PAGE analysis of CD protein.** Transformed BL21 cells induced with IPTG were separated into soluble and insoluble fractions and subjected to SDS-PAGE. M, molecular weight marker (kDa); 1, total cell before induction; 2, total cell after induction; 3, total cell after sonication; 4, supernatant; 5, pellet solubilized in 8 M urea. **(c) Chromatogram showing the elution profile of CD.** The denatured insoluble fraction was subjected to protein purification using a Ni-Sepharose High performance (NiHP) affinity column on an AKTA pure™ chromatography system. Fraction numbers 8–10 were pooled. **(d) SDS-PAGE analysis of CD protein during purification.** The proteins were visualized by Coomassie staining. M, molecular weight marker (kDa); 1, sample before loading on the column; 2, flow-through; 3–7, elution fractions after NiHP chromatography. **(e) SDS-PAGE analysis of the expressed polypeptides.** BL21 cells transformed with plasmids carrying the respective constructs were induced with IPTG, and the extracts were subjected to SDS-PAGE under reducing conditions, followed by Coomassie staining. 1, Polypeptide A; M, molecular weight marker (kDa); 2, Polypeptide CD; 3, Polypeptide E; 4, Polypeptide FG; 5, MVM VP2 protein; 6, MVM NS1 protein; 7, KRV VP2 protein, 8, KRV NS1 protein. **(f) Western blotting.** Proteins transferred onto PVDF membrane were probed using HRP-conjugated anti-His MAb. 1, Polypeptide A; M, molecular weight marker (kDa); 2, Polypeptide CD; 3, Polypeptide E; 4, Polypeptide FG; 5, MVM VP2 protein; 6, MVM NS1 protein; 7, KRV VP2 protein, 8, KRV NS1 protein. The position of three molecular weight markers (kDa) is shown on the left.

**Table 3. Protein yield.**

| Protein designation | Predicted size (kDa) | Final yield after dialysis (mg/L) |
|---|---|---|
| A | 28.4 | 7.0 |
| CD | 42.9 | 20.0 |
| E | 31.9 | 2.2 |
| FG | 49.3 | 3.0 |
| MVM VP2 FL | 68.6 | 3.5 |
| MVM NS1 FL | 79.5 | 2.0 |
| KRV VP2 FL | 64.3 | 4.5 |
| KRV NS1 FL | 75.9 | 5.5 |

MVM and KRV), were tested for reactivity against sera previously identified as positive or negative with a commercial kit, to evaluate the suitability of such constructs in ELISA.

For both MVM and KRV, full-length VP2 or NS1, a fragment of VP2 or NS1 containing antigenic clusters, and a fragment which is a fusion of antigenic clusters of NS1 (Fig 1) were amplified by either conventional PCR or by SOE-PCR (Fig 2a). Fragments A and E were PCR amplified and cloned into pRSETA vector. The CD and FG fragments were obtained by joining C with D (Figs 1 and 2a) and F with G, with a five amino acid linker between them (Fig 1).

All the recombinant proteins [MVM VP2 (predicted to be 68.6 kDa), MVM NS1 (79.5 kDa), A (28.4 kDa), CD (42.9 kDa), KRV VP2 (64.3 kDa), KRV NS1 (75.9 kDa), E (31.9 kDa) and FG (49.3 kDa)] were expressed in BL21 (λDE3) cells, using IPTG induction. For example, the expression, localization, and purification profile of CD protein is shown (Fig 2). The desired sized band (42.9 kDa) was observed upon induction with IPTG (Fig 2b, lane 2), and after sonication, the protein was localized majorly in the pellet (Fig 2b, lane 5). The pellet was solubilized in 8 M urea and purified by using the Ni-affinity column under denaturing conditions (Fig 2c). Peak elution fractions (such as 5 and 6 in Fig 2d) were pooled and analyzed by SDS-PAGE and Western blotting. Fig 2e represents the SDS-PAGE analysis of induced cell lysates for different proteins. All the proteins were purified similarly and dialyzed. The yield of proteins ranged from 2 mg to 20 mg per L of culture (Table 3). Western blotting with anti-His antibodies confirmed the presence of recombinant proteins in the preparations (Fig 2f).

A checkerboard titration was performed to standardize different parameters and to obtain good segregation in OD values between positive and negative samples. Of the 130 sera characterized using the commercial MVM ELISA kit (112 negative and 18 positive), six negative and 14 positive sera were chosen for optimization of in-house ELISA based on the availability of sufficient volumes for subsequent analysis. Similarly, of the 126 sera characterized using the commercial KRV ELISA kit (116 negative and ten positive), 14 negative and four positive sera were chosen. As antigen, individual proteins as well as a mixture of two proteins were tested, and coating concentrations were optimized for MVM VP2 (5 μg/ml), MVM NS1 (5 μg/ml), A (5 μg/ml), CD (5 μg/ml), KRV VP2 (2 μg/ml), KRV NS1 (5 μg/ml), E (3 μg/ml) and FG (5 μg/ml) and combination of two proteins (2.5 μg/ml each). The ideal dilution of the test sera was 1:100, and that of the HRP-conjugated anti-mouse or anti-rat IgG (H+L) was 1:10,000. The cut-off value was determined by calculating the average OD of negative samples + 3*SD.

The evaluation of in-house MVM/KRV ELISA against the commercial kits is shown in Fig 3 for three representative samples. A combination of A and NS1 was the best antigen for detecting antibodies against MVM, whereas, for KRV, all the antigen preparations worked fairly equally well.

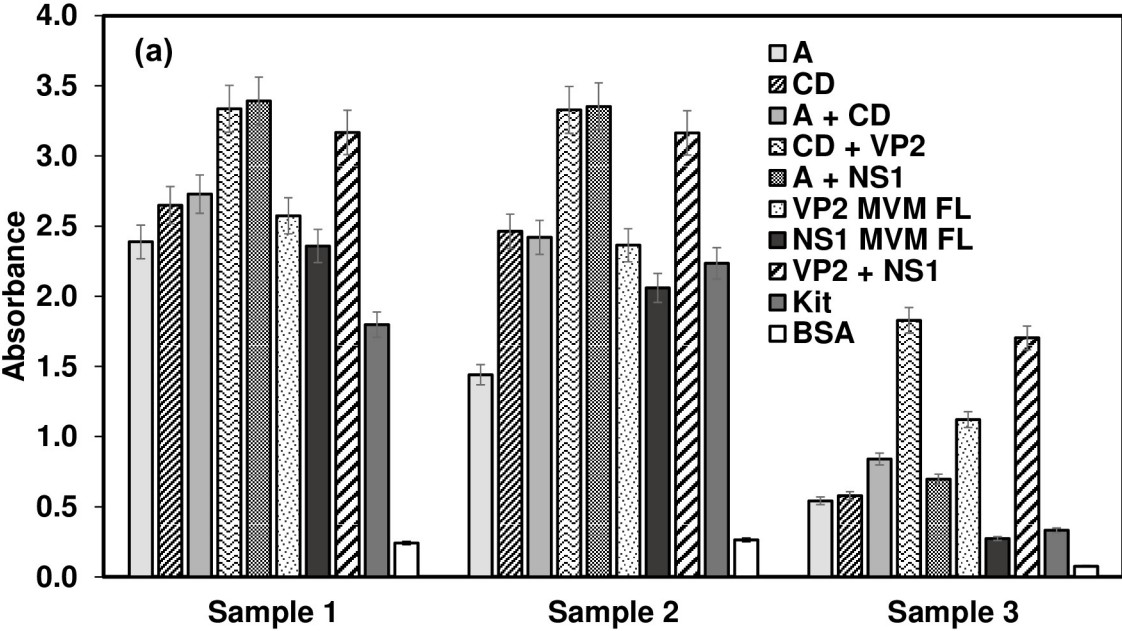

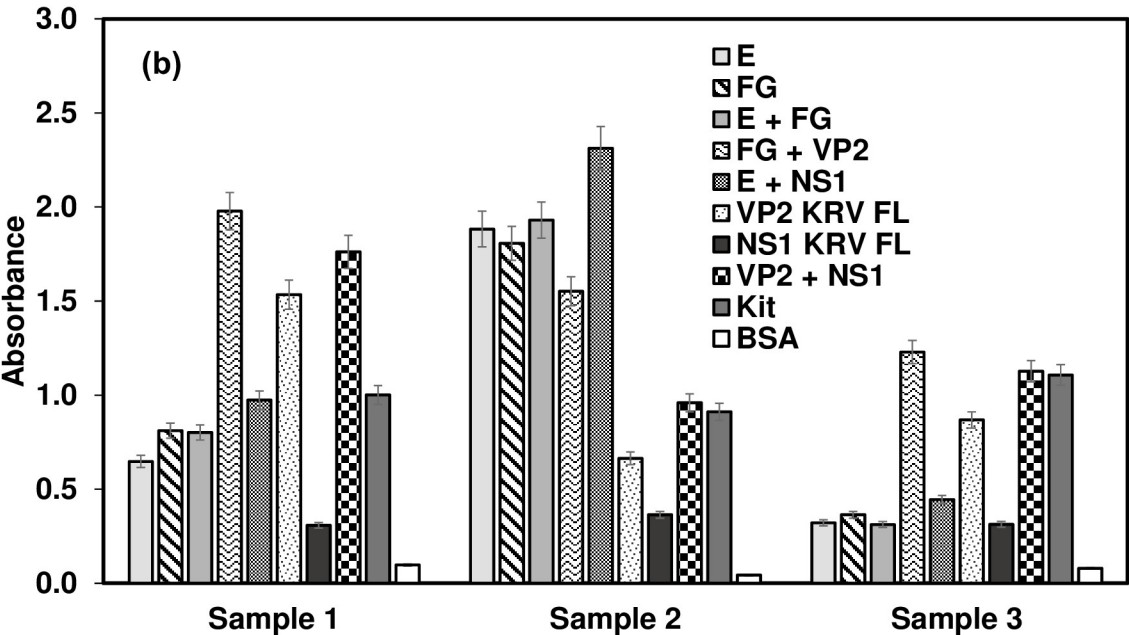

**Fig 3. Comparison of in-house ELISA with Xpress Bio kit.** Wells were coated with a predetermined quantity of the polypeptides, blocked, and reacted with serum samples before detecting with HRP-conjugated anti-mouse or anti-rat antibodies. BSA was used as the control antigen to deduct the background OD. Representative data for three mouse (a) and three rat (b) serum samples are shown respectively for MVM and KRV.

**Table 4. Performance of in-house ELISAs against commercial ELISA kits.**

|  | MVM | | KRV | |
|---|---|---|---|---|
|  | Kit | In-house ELISA | Kit | In-house ELISA |
| No. of samples analyzed | 20 | 20 | 18 | 18 |
| Positive | 14 | 18 | 4 | 9 |
| Negative | 6 | 2 | 14 | 9 |

The in-house ELISA detected the negative and positive control sera congruent with the commercial kit. However, the in-house ELISA produced signals greater than cut-off values for an additional four samples compared to the commercial kit (Table 4; S1 Table). These four samples were positive with the fragments as well as with the full-length proteins. A mixture of CD and VP2 worked best to detect antibodies against MVM (Fig 4a). A combination of A and CD for coating worked better than individually coated A or CD proteins to detect antibodies in 90% of the serum samples (Fig 3a, samples 1 and 3). The VP2 FL alone and a combination of VP2 and NS1 also detected 18 samples positive out of 20 (Table 4; S1 Table); a synergistic effect was observed using a combination of both VP2 and NS1 protein (Figs 3a and 4a). Most significant results were obtained with full-length VP2 combined with either the full-length NS1 or fragment CD (Table 5).

For KRV ELISA, out of the18 samples tested, the in-house ELISA was able to detect nine positive samples as compared to 4 samples detected positive by the kit (Table 4; S1 Table). The additional five samples were positive with fragments as well as the full-length proteins. A combination of E and FG and FG and NS1 worked best to detect anti-KRV antibodies (Figs 3b and 4b; Table 4; S1 Table). Synergistic signal enhancement was observed due to synergistic effect when using a combination of proteins for coating (Fig 4). Most significant results were obtained with a combination of full-length NS1 with the fragment E (Table 6).

## Discussion

We describe the design and evaluation of potential antigenic regions of VP2 and NS1 proteins of MVM and KRV for assessing their performance in serodiagnosis. The multiepitope fragment of VP2 and the predicted immunodominant regions of NS1 stitched together as one recombinant protein were cloned, expressed, purified, and evaluated for their ability to detect antibodies in the sera samples by indirect ELISA. The performance of our in-house assay was compared with commercial kits that use inactivated viruses. The in-house ELISA detected 20% more sera as positive for MVM and 27% more of the sera as positive for KRV. Notably, the same result was obtained with more than one construct for both MVM and KRV, suggesting that these results are genuine.

Conventionally, inactivated viruses are used for the detection of specific antibodies. However, this necessitates the propagation and purification of viruses, which in turn requires higher biocontainment than that required for the production of prokaryotically expressed protein antigens. In addition, in some cases, kits using inactivated viruses have been reported to be less than satisfactory [33, 35]. Recombinant proteins or virus-like particles have been increasingly proposed for use in the development of diagnostic assays [31, 33]. Our study supports previous reports applying full-length VP2 and NS1 to detect anti-MVM/KRV antibodies in mice/rat sera. In one study, recombinant MVM VP2 was expressed to produce virus-like particles in a baculovirus-infected insect cell system [31]. However, this system needed purification by CsCl gradient ultracentrifugation, which is a disadvantage in large-scale production, mainly because the final preparation may also contain baculovirus virions or sub-virion

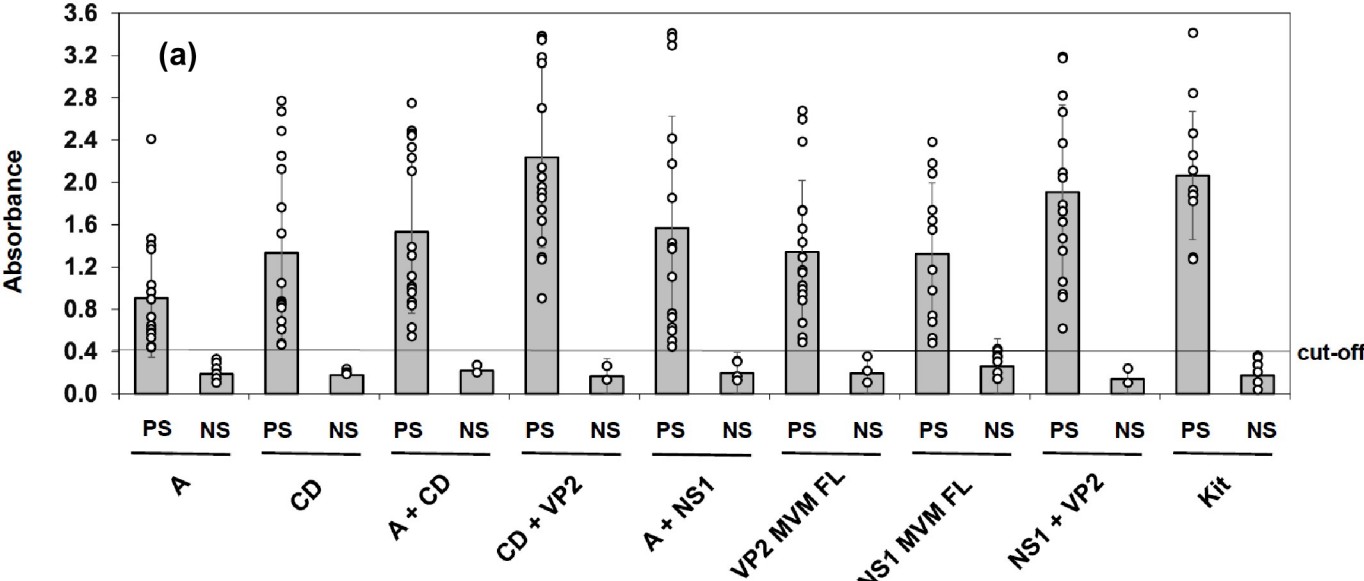

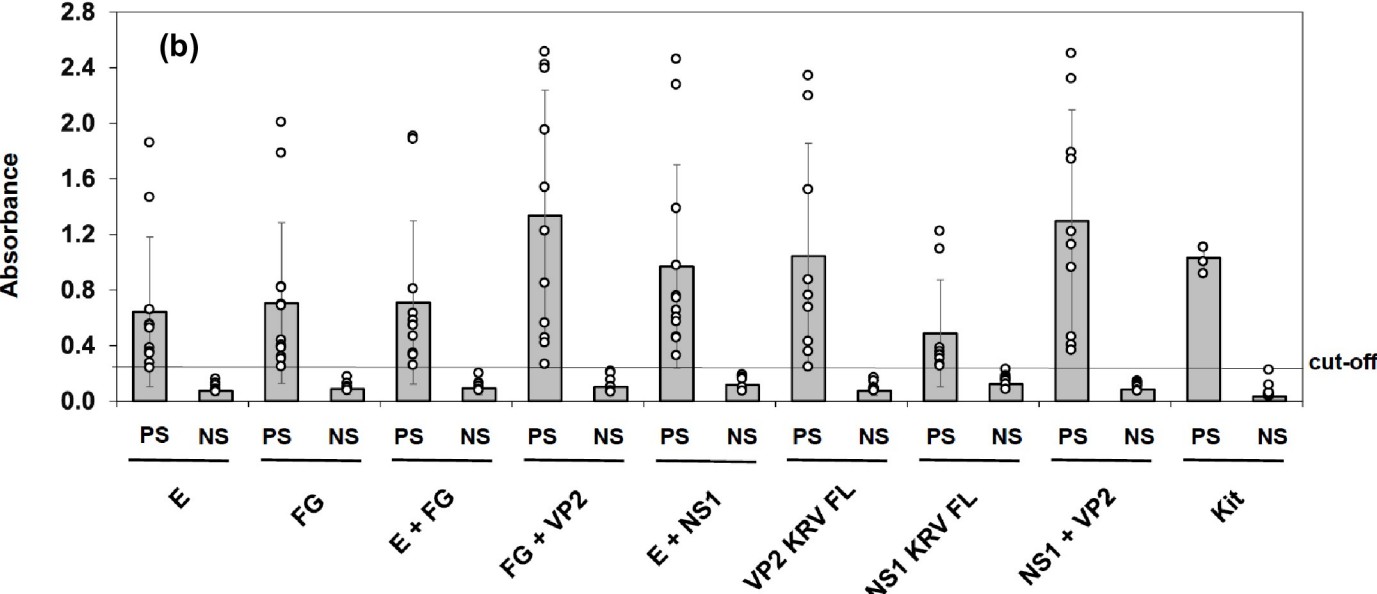

**Fig 4. Scatter plots showing the reactivity or sera against various polypeptides.** OD values for the positive (PS) and negative (NS) serum samples were plotted against each of the polypeptides used in the study. Data for mouse sera against MVM (a) and rat sera against KRV (b) are shown separately. PS and NS denote positive and negative sera, respectively, as determined using the commercial kit.

particles. Another study reported the development of an indirect ELISA employing MVM produced in BHK-21 cells as an antigen [29], but this has the limitation of the requirement of cell culture and virology methods followed by purification of the virus/antigen.

The efficiency of immobilizing a mixture of proteins in ELISA has been evaluated using mouse monoclonal or rabbit polyclonal antibodies [36]. In some cases, testing against multiple

**Table 5. P-values for significance of difference for various antigens used for MVM ELISA.**

|  | A | CD | A + CD | CD + VP2 | A + NS1 | VP2 FL | NS1 FL | NS1 + VP2 |
|---|---|---|---|---|---|---|---|---|
| **Kit** | **0.0128** | 0.3925 | 0.8721 | **0.0005** | 0.9510 | 0.5180 | **0.0220** | **0.0377** |
| **A** |  | **0.0022** | **0.0001** | **<0.0001** | **0.0005** | **0.0001** | 0.0630 | **<0.0001** |
| **CD** |  |  | **0.0033** | **<0.0001** | 0.3318 | 0.9342 | **0.0121** | **0.0001** |
| **A + CD** |  |  |  | **<0.0001** | 0.7811 | 0.2451 | **0.0006** | **0.0044** |
| **CD + VP2** |  |  |  |  | **0.0001** | **<0.0001** | **<0.0001** | **0.0093** |
| **A + NS1** |  |  |  |  |  | 0.3892 | **0.0010** | **0.0011** |
| **VP2 FL** |  |  |  |  |  |  | **0.0193** | **<0.0001** |
| **NS1 FL** |  |  |  |  |  |  |  | **<0.0001** |

proteins seems necessary to achieve a desired sensitivity range [37–39]. Nevertheless, multi-epitope proteins or a fusion of multiple proteins are beneficial for multiplexing to achieve desired assay sensitivity or to detect many serotypes [40–42]. However, such proteins are challenging to produce due to solubility issues. The utility of a mixture of proteins and/or multi-epitope fragments as capture molecules described here may facilitate several applications, including the development of multiplexed immunoassays for antibody detection in serum samples.

Ultimately, the requirement for any test depends on its fit for purpose. A diagnostic test requires higher specificity to rule in a disease, whereas a screening test requires high sensitivity to rule out a disease. In the context of laboratory animals, an experimental set-up would require the detection of a specific pathogen being studied whereas health monitoring would involve screening for the presence of a list of pathogens in a colony. A test with both high sensitivity and specificity could be applied to both situations. Notably, we used a serum dilution of 1:100 to derive the cut-off value for declaring a sample as positive or negative. And yet, more samples were declared positive with our assay than the commercial kit, allowing the tests to be performed with a lesser volume (presumably ten times less) of serum, which would be very beneficial, particularly in cases where repeated sampling is planned as per the experimental outlay. On the other hand, sample volume may not be a limitation for sentinel screening.

The present manuscript describes a proof-of-concept study, and does not aim to validate an already developed method. It is to be noted that we used previously collected samples for this study. Owing to limited number and quantity of sera, neither are the sera blinded nor the numbers sufficient to infer on the robustness of the assay. Our preliminary stidies are promising but require further studies with more number of sera to determine better the diagnostic sensitivity and specificity of the assay, and to rule out any cross-reactivities with other,

**Table 6. P-values for significance of difference for various antigens used for KRV ELISA.**

|  | E | FG | E + FG | FG + VP2 | E + NS1 | VP2 FL | NS1 FL | NS1 + VP2 |
|---|---|---|---|---|---|---|---|---|
| **Kit** | 0.1406 | 0.0600 | 0.0663 | **0.0030** | **0.0179** | **0.0339** | 0.5835 | **0.0059** |
| **E** |  | **0.0398** | **0.0060** | **0.0187** | **0.0020** | 0.4484 | 0.2926 | 0.0700 |
| **FG** |  |  | 0.3498 | 0.1073 | **0.0086** | 0.9108 | **0.0377** | 0.2692 |
| **E + FG** |  |  |  | 0.0537 | **0.0035** | 0.7795 | 0.0884 | 0.1755 |
| **FG + VP2** |  |  |  |  | 0.3441 | **0.0071** | **0.0130** | 0.0793 |
| **E + NS1** |  |  |  |  |  | 0.5489 | **0.0071** | 0.7039 |
| **VP2 FL** |  |  |  |  |  |  | 0.1880 | **0.0115** |
| **NS1 FL** |  |  |  |  |  |  |  | **0.0218** |

especially closely related pathogens. Further development and validation would require power analysis for sample size calculation as well as randomization and blinding. The challenge would be to achieve a sensitive, specific, easy-to-perform, and low-cost test that can be applied in resource-constrained settings.

## Supporting information

**S1 Fig. Raw images for gels and blots used to generate the figures.** The panels contain the original gels, with the lanes used for the figure being labeled with the same numbers, whereas the rest of the lanes are designated as 'x'.
(PDF)

**S1 Table. Summary of the results of in-house ELISA against Xpress Bio ELISA kit for the detection of antibodies to MVM.**
(PDF)

**S2 Table. Summary of the results of in-house ELISA against Xpress Bio ELISA kit for the detection of antibodies to KRV.**
(PDF)

## Acknowledgments

We are thankful to Dr. Girish Radhakrishnan, National Institute of Animal Biotechnology, for help with proofreading.

## Author Contributions

**Conceptualization:** Charanpreet Kaur, S. G. Ramachandra, Nagendra R. Hegde.

**Data curation:** Charanpreet Kaur, Kandala Pavan Asrith, Nagendra R. Hegde.

**Formal analysis:** Charanpreet Kaur, Nagendra R. Hegde.

**Funding acquisition:** S. G. Ramachandra, Nagendra R. Hegde.

**Investigation:** Charanpreet Kaur.

**Methodology:** Charanpreet Kaur, Kandala Pavan Asrith.

**Project administration:** S. G. Ramachandra, Nagendra R. Hegde.

**Resources:** S. G. Ramachandra, Nagendra R. Hegde.

**Supervision:** S. G. Ramachandra, Nagendra R. Hegde.

**Visualization:** Charanpreet Kaur.

**Writing – original draft:** Charanpreet Kaur, S. G. Ramachandra, Nagendra R. Hegde.

**Writing – review & editing:** Charanpreet Kaur, S. G. Ramachandra, Nagendra R. Hegde.

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
