## [Decision Letter · Decision Letter 0]

15 Oct 2023

PONE-D-23-29135Immunoinformatics-guided recombinant polypeptide-based enzyme-linked immunosorbent assay for seromonitoring of laboratory animals for minute virus of mice and Kilham rat virusPLOS ONE

Dear Dr. Hegde,

Thank you for submitting your manuscript to PLOS ONE. After careful consideration, we feel that it has merit but does not fully meet PLOS ONE’s publication criteria as it currently stands. Therefore, we invite you to submit a revised version of the manuscript that addresses the points raised during the review process.

We look forward to receiving your revised manuscript.

Kind regards,

Paulo Lee Ho, Ph.D.

Academic Editor

PLOS ONE

A clean copy of the edited manuscript (uploaded as the new *manuscript* file)”.

 [This study was supported by a grant (BT/PR15810/ADV/90/198/2015) from the Department of Biotechnology, Ministry of Science and Technology, Government of India, to NRH and SGR.].  

[This study was supported by a grant (BT/PR15810/ADV/90/198/2015) from the Department of Biotechnology, Ministry of Science and Technology, Government of India, to NRH and SGR.]

 [This study was supported by a grant (BT/PR15810/ADV/90/198/2015) from the Department of Biotechnology, Ministry of Science and Technology, Government of India, to NRH and SGR.]

Reviewers' comments:

Reviewer's Responses to Questions

**Comments to the Author**

1. Is the manuscript technically sound, and do the data support the conclusions?

Reviewer #1: Yes

Reviewer #2: Partly

2. Has the statistical analysis been performed appropriately and rigorously? 

Reviewer #1: I Don't Know

Reviewer #2: No

3. Have the authors made all data underlying the findings in their manuscript fully available?

Reviewer #1: No

Reviewer #2: Yes

4. Is the manuscript presented in an intelligible fashion and written in standard English?

Reviewer #1: Yes

Reviewer #2: Yes

5. Review Comments to the Author

Reviewer #1: The study employs immunoinformatics-driven polypeptide engineering to enhance the enzyme-linked immunosorbent assay (ELISA) method for detecting MVM and KRV infections in laboratory animals. The study demonstrates that the engineered fusion of fragments containing multiple predicted epitopes of MVM VP2 and NS1 exhibits superior sensitivity in detecting antibodies in serum samples compared to a commercial kit (Xpress Bio kit) that uses full-length VP2 and NS1 for the detection of anti-MVM/KRV antibodies in mice and rats, respectively.

The study represents a significant technical innovation regarding the sensitivity of the ELISA method. However, as acknowledged by the authors themselves, additional experiments are necessary to assess both sensitivity and, importantly, specificity. The increasing diversity of rodent parvoviruses and their impact on research underscores the necessity for specificity tests.

Another crucial consideration is the health status of the animals. Colonies maintained under specific-pathogen-free (SPF) conditions should be free of pathogens such as parvovirus. In this context, a more sensitive test may not significantly impact the outcome since detecting the virus in just a few sentinel animals (with no limitations on the quantity of samples) would suffice to trigger a report and initiate immediate colony cleaning procedures.

On the other hand, ongoing experiments could benefit from a more specific and sensitive ELISA test to identify infected animals, particularly in subclinical conditions. Depending on the expected impact on the results, this could lead to the elimination or treatment of affected animals.

The methodology for in silico prediction, cloning, expression, purification, and evaluation is well-organized and thoroughly documented. However, the performance was compared to only one commercial kit for each virus. It would be valuable to compare it to other commercial kits as well.

In line 278, FG + NS1 is not depicted in Fig. 3b and 4b, nor is it included in Table S1.

Basic statistical analyses are absent from Figures 3 and 4.

I recommend including specificity tests in the study.

Reviewer #2: The present study entitled “Immunoinformatics-guided recombinant polypeptide-based enzyme-linked immunosorbent assay for seromonitoring of laboratory animals for minute virus of mice and Kilham rat virus” aimed at the development of a novel ELISA test based immunoinformatics-driven polypeptide engineering.

The herein described assay is proposed to be beneficial compared to commercially available test kits for the detection of MVM and KRV specific antibodies, therefore being suggested as a fast and highly sensitive serodiagnostic method for the hygienic monitoring of laboratory mouse and rat colonies for parovirus infections.

Validation of the novel ELISA was performed re-using serum sample material from mice and rats, which was previously collected in various animal facilities during routine health monitoring procedures.

The manuscript is well written and structured, and the overall aim of the study becomes easily clear. However, critical reading of the article revealed some weaknesses, which I would like to specify in the following comments:

1) Although the study did not directly involve animal experimentation, the reporting should be generally based on the ARRIVE guidelines (Percie du Sert, N.; Hurst, V.; Ahluwalia, A.; Alam, S.; Avey, M.T.; Baker, M.; Browne, W.J.; Clark, A.; Cuthill, I.C.; Dirnagl, U.; et al.; The arrive guidelines 2.0: Updated guidelines for reporting animal research. PLoS Biol. 2020, 18, e3000410), since laboratory animal derived sample material was used.

The authors should here especially comment on all measures regarding sample size calculation, randomization of samples and blinding.

As this study aims to validate a diagnostic methods, inclusion and exclusion criteria of sample material should be discussed in detail and a statistical calculation of the results including estimation of the diagnostic sensitivity and specificity would be of value.

2) The authors used the novel ELISA on serum samples, which were re-used from a previous study. Can the authors comment on the storage conditions of the samples and potential influence on antibody stability?

The authors categorized samples in “positive” and “negative” based on results obtained by a commercial ELISA kit and re-tested them with the herein described ELISA. Since they tested four (mice) and 5 (rats) more samples positive compared to the commercial kit, they proposed that the novel ELISA would be “better” in terms of diagnostic sensitivity. How can the authors exclude, that the result from the novel method was false positive and lacks diagnostic specificity? Third-party testing (preferably with an alternative method) is strongly recommended to solve this discrepancy and draw proper conclusions. The number of analyzed negative samples should be generally increased to strengthen results to calculate the diagnostic test specificity (control for false positive test results).

Furthermore, the results part should include a proper calculation of the diagnostic sensitivity and specificity of the assay.

3) Line 342: reference 4 – Mähler, not Mahler

6. PLOS authors have the option to publish the peer review history of their article (what does this mean?). If published, this will include your full peer review and any attached files.

Reviewer #1: No

Reviewer #2: No

---

## [Author Response · Author response to Decision Letter 0]

7 Dec 2023

Responses to Reviewers

Reviewer #1

1. The study represents a significant technical innovation regarding the sensitivity of the ELISA method. However, as acknowledged by the authors themselves, additional experiments are necessary to assess both sensitivity and, importantly, specificity. The increasing diversity of rodent parvoviruses and their impact on research underscores the necessity for specificity tests.

Another crucial consideration is the health status of the animals. Colonies maintained under specific-pathogen-free (SPF) conditions should be free of pathogens such as parvovirus. In this context, a more sensitive test may not significantly impact the outcome since detecting the virus in just a few sentinel animals (with no limitations on the quantity of samples) would suffice to trigger a report and initiate immediate colony cleaning procedures.

On the other hand, ongoing experiments could benefit from a more specific and sensitive ELISA test to identify infected animals, particularly in subclinical conditions. Depending on the expected impact on the results, this could lead to the elimination or treatment of affected animals.

Response: We deeply appreciate these comments from the Reviewer. We have included some of this information in the penultimate paragraph of Discussion.

2. The methodology for in silico prediction, cloning, expression, purification, and evaluation is well-organized and thoroughly documented. However, the performance was compared to only one commercial kit for each virus. It would be valuable to compare it to other commercial kits as well.

Response: Comparison to only one commercial kit was due to the non-availability of other kits in India at the time the study was conducted. In addition, we had limitations on funds.

3. In line 278, FG + NS1 is not depicted in Fig. 3b and 4b, nor is it included in Table S1.

Response: We regret that there was an error in both the Figure and the Table. It should have been FG + VP2 FL and E + NS1 Fl. The fragment FG is part of NS1 itself, and hence tests have not been performed with FG + NS1 combination. These have now been corrected.

4. Basic statistical analyses are absent from Figures 3 and 4.

Response: We thank the reviewer for pointing this out. We are of the opinion that statistical analysis is inappropriate for individual samples as depicted in Fig. 3. However, analysis of values against that of the kit for the samples shown in Fig. 3a and 3b is now stated in the text.

On the other hand, significance analyses are now provided for various comparisons for Fig. 4a and 4b, where all samples are tested against the various antigens or combinations. Two tables showing the P values for the various comparisons are now included.

5. I recommend including specificity tests in the study.

Response: It is not clear to us whether the Reviewer means cross-reactivity or diagnostic specificity here. If it was the former, such tests are difficult to do without pathogen-specific antigens and antisera. While it may be possible to procure antisera, it is difficult to obtain pathogen-specific antigens, and in particular, recombinant antigens that are known to work in the format that we have applied. If the Reviewer was referring to diagnostic specificity, we would like to submit that the manuscript describes proof-of-concept and requires much more work to evaluate and validate diagnostic parameters (see response to Reviewer #2).

Reviewer #2

1. Although the study did not directly involve animal experimentation, the reporting should be generally based on the ARRIVE guidelines (Percie du Sert, N.; Hurst, V.; Ahluwalia, A.; Alam, S.; Avey, M.T.; Baker, M.; Browne, W.J.; Clark, A.; Cuthill, I.C.; Dirnagl, U.; et al.; The arrive guidelines 2.0: Updated guidelines for reporting animal research. PLoS Biol. 2020, 18, e3000410), since laboratory animal derived sample material was used.

The authors should here especially comment on all measures regarding sample size calculation, randomization of samples and blinding.

As this study aims to validate a diagnostic methods, inclusion and exclusion criteria of sample material should be discussed in detail and a statistical calculation of the results including estimation of the diagnostic sensitivity and specificity would be of value.

Response: We have considered reporting based on ARRIVE guidelines while formulating the manuscript. However, we would like to submit that this is a proof-of-concept study. In addition, ARRIVE guidelines are not mandatory to be followed in India, particularly for experimental research. Furthermore, the samples had been collected during an earlier study for the purpose of routine surveillance for pathogens at various laboratory animal colonies in India.

We would like to submit that our study is only a proof-of-concept study, and does not aim to validate an already developed method. Further development and validation would require power analysis for sample size calculation as well as randomization and blinding. Unfortunately, there is not enough data on the prevalence of laboratory animal pathogens in India to do a power analysis and calculate the sample size. Also, the study was conducted by using previously collected samples and hence it lacks randomization and blinding. This is noted in the last paragraph of Discussion as a limitation. 

2. The authors used the novel ELISA on serum samples, which were re-used from a previous study. Can the authors comment on the storage conditions of the samples and potential influence on antibody stability?

Response: The samples had been stored at -80oC with very little freeze-thaw, and hence any instability of the antibodies can be ruled out.

3. The authors categorized samples in “positive” and “negative” based on results obtained by a commercial ELISA kit and re-tested them with the herein described ELISA. Since they tested four (mice) and 5 (rats) more samples positive compared to the commercial kit, they proposed that the novel ELISA would be “better” in terms of diagnostic sensitivity. How can the authors exclude, that the result from the novel method was false positive and lacks diagnostic specificity? Third-party testing (preferably with an alternative method) is strongly recommended to solve this discrepancy and draw proper conclusions. The number of analyzed negative samples should be generally increased to strengthen results to calculate the diagnostic test specificity (control for false positive test results).

Furthermore, the results part should include a proper calculation of the diagnostic sensitivity and specificity of the assay.

Response: We completely agree with the Reviewer. We had been careful not to use the word ‘better’ in comparison with the commercial kit, and only stated that more samples were positive. We have still scrutinized the whole manuscript and avoided the use of the word ‘better.’ Also see response to #1 above.

As far as diagnostic sensitivity and specificity is concerned, and as stated above, these cannot be determined with this proof-of-concept study where sufficient number of samples were not used.

4. Line 342: reference 4 – Mähler, not Mahler

Response: We thank the reviewer for pointing this out. It has now been changed.

---

## [Decision Letter · Decision Letter 1]

14 Dec 2023

PONE-D-23-29135R1Immunoinformatics-guided recombinant polypeptide-based enzyme-linked immunosorbent assay for seromonitoring of laboratory animals for minute virus of mice and Kilham rat virusPLOS ONE

Dear Dr. Hegde,

Thank you for submitting your manuscript to PLOS ONE. After careful consideration, we feel that it has merit but does not fully meet PLOS ONE’s publication criteria as it currently stands. Therefore, we invite you to submit a revised version of the manuscript that addresses the points raised during the review process. In addition, we authors should include in the manuscrit that the present manuscript is a proof-of-concept study, and does not aim to validate an already developed method. Further development and validation would require power analysis for sample size calculation as well as randomization and blinding. 

We look forward to receiving your revised manuscript.

Kind regards,

Paulo Lee Ho, Ph.D.

Academic Editor

PLOS ONE

Journal Requirements:

Reviewers' comments:

Reviewer's Responses to Questions

**Comments to the Author**

1. If the authors have adequately addressed your comments raised in a previous round of review and you feel that this manuscript is now acceptable for publication, you may indicate that here to bypass the “Comments to the Author” section, enter your conflict of interest statement in the “Confidential to Editor” section, and submit your "Accept" recommendation.

Reviewer #2: All comments have been addressed

2. Is the manuscript technically sound, and do the data support the conclusions?

Reviewer #2: Yes

3. Has the statistical analysis been performed appropriately and rigorously? 

Reviewer #2: Yes

4. Have the authors made all data underlying the findings in their manuscript fully available?

Reviewer #2: Yes

5. Is the manuscript presented in an intelligible fashion and written in standard English?

Reviewer #2: Yes

6. Review Comments to the Author

Reviewer #2: Thank you very much for adressing all comments and making the changes of the manuscript accordingly.

- Consider adding the statistical analysis also in the figure legend of Fig. 3

- "Table 5" is used for caption twice (one for p values of MVM and one for KRV)

- line 386 -392: Consider rephrasing

- line 401: To to

7. PLOS authors have the option to publish the peer review history of their article (what does this mean?). If published, this will include your full peer review and any attached files.

Reviewer #2: No

---

## [Author Response · Author response to Decision Letter 1]

27 Dec 2023

We once again thank the reviewer and the editor for the comments. We have provided point-by-point responses below, and have made the necessary changes in the manuscript, including some other minor errors that we spotted during this revision.

Response to Editor’s comments

1. Authors should include in the manuscript that the present manuscript is a proof-of-concept study, and does not aim to validate an already developed method. Further development and validation would require power analysis for sample size calculation as well as randomization and blinding.

Response: This has been included in the last paragraph of the Discussion. This paragraph has also been revised for coherence and to avoid repetition of information.

Response to Journal’s comments

Response: All the references are correct and none of them has been retracted.

Responses to Reviewers

Reviewer #2

1. Consider adding the statistical analysis also in the figure legend of Fig. 3.

Response: We thank the reviewer for this comment. Indeed, we considered adding statistical analysis for these samples while revising the manuscript which was submitted last. However, we feel that better inference can be drawn from statistical analysis on samples from a group of animals, particularly for diagnostic assays. It may be noted that analysis for individual samples may be more suited, for instance, for analysing immunogenicity of individual animals against an antigen. In order to keep the inference simple and meaningful in the context of a group of samples, we would not like to add statistical analysis for the data in Fig. 3.

2. "Table 5" is used for caption twice (one for p values of MVM and one for KRV).

Response: We apologize. It should have been Table 5 and Table 6. This has now been corrected both in the text and the table captions.

3. line 386 -392: Consider rephrasing

Response: We are unable to figure out what this refers to as there seems to be some discrepancy in the line numbers in document that the reviewer is seeing and the pdf document that was generated by the system. For example, line 401 as noted below is a blank line. However, we have rephrased what we think is a bit confusing and hope that it is clearer now. This is the second sentence in the 4th paragraph in the Discussion (lines 374-376).

4. line 401: To to

Response: Thank you; this has been corrected. This in the 4th line of the last paragraph of the Discussion (line 386).

---

## [Decision Letter · Decision Letter 2]

30 Jan 2024

Immunoinformatics-guided recombinant polypeptide-based enzyme-linked immunosorbent assay for seromonitoring of laboratory animals for minute virus of mice and Kilham rat virus

PONE-D-23-29135R2

Dear Dr. Hegde,

We’re pleased to inform you that your manuscript has been judged scientifically suitable for publication and will be formally accepted for publication once it meets all outstanding technical requirements.

Kind regards,

Paulo Lee Ho, Ph.D.

Academic Editor

PLOS ONE

Additional Editor Comments (optional):

Reviewers' comments:

Reviewer's Responses to Questions

**Comments to the Author**

1. If the authors have adequately addressed your comments raised in a previous round of review and you feel that this manuscript is now acceptable for publication, you may indicate that here to bypass the “Comments to the Author” section, enter your conflict of interest statement in the “Confidential to Editor” section, and submit your "Accept" recommendation.

Reviewer #2: All comments have been addressed

2. Is the manuscript technically sound, and do the data support the conclusions?

Reviewer #2: Yes

3. Has the statistical analysis been performed appropriately and rigorously? 

Reviewer #2: Yes

4. Have the authors made all data underlying the findings in their manuscript fully available?

Reviewer #2: Yes

5. Is the manuscript presented in an intelligible fashion and written in standard English?

Reviewer #2: Yes

6. Review Comments to the Author

Reviewer #2: Thank you very much for adressing all points raised throughout the reviewing procedure. I have no further comments.

7. PLOS authors have the option to publish the peer review history of their article (what does this mean?). If published, this will include your full peer review and any attached files.

Reviewer #2: No

---

## [Editor Report · Acceptance letter]

17 Feb 2024

PONE-D-23-29135R2 

PLOS ONE

Dear Dr. Hegde, 

I'm pleased to inform you that your manuscript has been deemed suitable for publication in PLOS ONE. Congratulations! Your manuscript is now being handed over to our production team.

Kind regards, 

on behalf of

Dr. Paulo Lee Ho 

Academic Editor

PLOS ONE